# Catechins and Proanthocyanidins Involvement in Metabolic Syndrome

**DOI:** 10.3390/ijms24119228

**Published:** 2023-05-25

**Authors:** Giuseppe Tancredi Patanè, Stefano Putaggio, Ester Tellone, Davide Barreca, Silvana Ficarra, Carlo Maffei, Antonella Calderaro, Giuseppina Laganà

**Affiliations:** Department of Chemical, Biological, Pharmaceutical and Environmental Sciences, University of Messina, Viale Ferdinando Stagno d’Alcontres 31, 98166 Messina, Italy; giuseppe.patane@studenti.unime.it (G.T.P.); stefano.putaggio@studenti.unime.it (S.P.); silvana.ficarra@unime.it (S.F.); carlo.maffei@studenti.unime.it (C.M.); anto.calderaro@gmail.com (A.C.); giuseppina.lagana@unime.it (G.L.)

**Keywords:** cardiovascular diseases, natural antioxidants, inflammatory state, flavan-3-ols, catechins, proanthocyanidins

## Abstract

Recent studies on natural antioxidant compounds have highlighted their potentiality against various pathological conditions. The present review aims to selectively evaluate the benefits of catechins and their polymeric structure on metabolic syndrome, a common disorder characterized by a cluster of three main risk factors: obesity, hypertension, and hyperglycemia. Patients with metabolic syndrome suffer chronic low inflammation state and oxidative stress both conditions effectively countered by flavanols and their polymers. The mechanism behind the activity of these molecules has been highlighted and correlated with the characteristic features present on their basic flavonoidic skelethon, as well as the efficient doses needed to perform their activity in both in vitro and in vivo studies. The amount of evidence provided in this review offers a starting point for flavanol dietary supplementation as a potential strategy to counteract several metabolic targets associated with metabolic syndrome and suggests a key role of albumin as flavanol-delivery system to the different target of action inside the organism.

## 1. Introduction

Recent studies on natural antioxidant compounds have highlighted their great potentiality against several pathological conditions associated with metabolic and physiological disorders referred to as metabolic syndrome (MetS). Among these compounds, flavonoids belonging to the group of polyphenols, are found in many foods, such as vegetables, fruits, tea and in flowers and wine [1]. More than 10,000 compounds have been found and this makes the group quite heterogeneous and divisible into further categories and subgroups [2]. Some of these compounds are also responsible for the color of fruits, vegetables, and flowers as well as the yellow and orange color of citrus fruits and the red and blue color of berries [3]. Many experimental studies describe different roles and biological activities of flavonoids. Different epidemiological studies have evidenced that they have a pivotal role as far as the prevention of cardiovascular diseases and the onset of the corresponding pathological conditions are concerned [4]. These compounds, also called bioflavonoids, have had growing interest because of their multiple beneficial effects and multitarget action; their antimicrobial, anti-inflammatory, antiaggregating, antiamylogenic, and strong antioxidant activities have been demonstrated. Furthermore, many of them may exert positive health effects on central nervous system, such as antioxidant capacity, thanks to their ability to cross the blood–brain barrier [4,5]. Biological evidence and their abundance in the Mediterranean diet make flavonoids a promising strategy to counteract the development of MetS and lower the risks associated with it [6]. Since there are different subgroups of flavonoids with different biological activities depending on the chemical structure and substituents, the present review aims to selectively evaluate the benefits of some categories of flavanol and their polymeric structure on MetS.

## 2. Metabolic Syndrome (MetS)

MetS is a general term that includes a cluster of metabolic abnormalities such as insulin resistance, central obesity, hypertension, and atherogenic dyslipidemia. In fact, it is also known as ‘insulin resistance syndrome’, ‘syndrome X’, ‘hypertriglyceridemic waist’, and ‘the deadly quartet’. Based on the definition of the World Health Organization (although this definition has widely integrated and modified), MetS occurs when, in the presence of insulin resistance, there is the addition of other two risk factors such as obesity, hypertension, hyperlipidemia, or microalbuminuria. In each case, it is also strongly associated with a potential increased risk factor for developing diabetes and atherosclerotic and cardiovascular disease (CVD) [7]. It is a pathology linked to both genetic and acquired factors that contribute to bringing about an inflammation state that is one of the main responsible for the development of CVD. Moreover, the last twenty years have been characterized by a dramatic increase in obesity with a serious impact on public health. The excess of adipose tissue represents an important risk factor for health as it induces the establishment of a low-grade inflammatory state, silent but chronic. This condition hinders weight loss and is associated with multiple clinical complications, including hypertension, dyslipidemia, insulin resistance, high levels of cholesterol, and triglycerides in the blood and in severe cases, cardiovascular disease, type two diabetes, and nonalcoholic fatty liver disease (NAFLD). Especially, the abdominal distribution of adipose tissue is of great importance regarding these comorbidities known as metabolic syndrome [8]. The diagnosis of MetS focuses on the assessment at least three of the following risk factors visceral fat accumulation, hypertension, hyperglycemia, and dyslipidemia [9]. People affected by this condition are at high risk of developing type 2 diabetes and are twice as likely to develop coronary heart disease [10]. In addition, a wide range of other disorders occur concomitantly with or because of MetS, including NAFLD. The pathophysiology of MetS is complex; in general, it is characterized by a state of neurohormonal activation; insulin resistance and release of fatty acids from adipose tissue; and an increase in oxidative stress accompanied by a low-grade inflammation [11,12,13]. Patients with MetS are characterized by chronic systemic inflammation, a process related to oxidative stress and genetic mutations; several studies have shown a close correlation between the progression of the dysmetabolic process and an altered increase in the release of adipokines including tumor necrosis factor alpha (TNF a) and interleukin-6 (IL-6) linked with a progressive activation of the innate immune system [14,15,16]. An increased production of reactive oxygen (ROS) and nitrogen (RNS) species caused by the imbalance between pro-oxidizing and antioxidants is another distinctive feature of MetS [17]. Reduced catalytic activity of catalase (CAT), superoxide dismutase (SOD), and nitric oxide endothelial synthase (eNOS) and decreased levels of glutathione peroxidase (GPx) and glutathione-S-transferase (GST) were detected in plasma of patients with MetS compared to healthy individuals [17,18]. On the contrary, the myeloperoxidase activity was higher. The antioxidant, antihyperlipidemic, and anti-inflammatory properties of polyphenols could help to counteract the main risk factors characterizing the onset and establishment of a dysmetabolic state. In fact, although cells have a valid endogenous antioxidant system equipped with enzymatic scavengers of ROS, these defenses are not always able to efficiently counteract the enhanced ROS production. Therefore, agents that can strengthen antioxidant defense and prevent the increase in ROS generation may represent an effective treatment to combat inflammation-related oxidative stress in the development of numerous diseases.

## 3. Chemical Structure and Classification of Flavonoids

Flavonoids belong to the group of polyphenols, as they are secondary metabolites of plants [19]. From a chemical point of view, all these compounds are characterized by a basic nucleus containing 15 carbon atoms, in which the scaffold is represented by a 2-phenylbenzo-γ-pyron (Figure 1). These carbon atoms are distributed in such a way as to form two six-term aromatic rings, commonly denoted by “A” and “B”, connected by a three-carbon bridge that usually cyclizes through an oxygen atom to form a third pyron ring “C” connecting the first two aromatic rings [20]. Based on the vastness of the compounds identified over time, flavonoids may be divided into different groups. Following a generic classification, based on different characteristics, such as the number of hydroxyl groups, the type of substituents, the degree of unsaturation of the central ring, and the oxidation of carbon atoms, it is possible to recognize three subclasses: bioflavonoids, isoflavonoids, and neoflavonoids [21].

According to chemical structure, a more detailed classification allows to identify six subgroups, which are flavones, flavanones, flavonols, isoflavonoids, anthocyanidins, and flavanols [22]. Catechins, in particular, is a subgroup of flavonols that shows the basic characteristic of this group of compounds. Since the purpose of our review is to analyze only the effects of catechins and their polymers, we will focus in more detail on the chemical characteristics of the following categories. In the flavan-3-ols, it is in fact possible to observe the presence of 2 benzene rings (indicated with the letters A and B) and a dihydropyran heterocyclic ring (C), with a -OH (hydroxyl group) bound to C3 [23]. The basic skeleton is particularly rich in hydroxyls groups and, based on the location of these substituents on the A ring, we can divide flavan-3-ols into two categories: meta-dihydroxylated or meta-trihydroxylated compounds and, according to the hydroxylation on the B ring, flavan-3ols monohydroxylates, orthohydroxylates, or vicinal-hydroxylates compounds [24]. Another important chemical feature of these compounds is that they have three chiral centers in position C2, C3, and C4 on the heterocyclic ring; as a result, different stereoisomers are possible (Figure 1). Since the R configuration on the C2 is almost exclusively present in nature, it was originally assumed that this was the only possible configuration; nevertheless, flavanols with opposite configuration on C2 (S) have been found [25]. When on C2 and C3 substituents are oriented in the same direction, the prefix epi is added, to obtain (+) catechin (2S, 3R) and (−) epi (2R,3R) [26]. A peculiarity of catechins is the ability to form oligodimers or high molecular weight polymers—called proanthocyanidins (PCAs) or condensed tannins—through polymerization reactions that mainly involve catechin (C), gallocatechin (EC), epicatechin (EC), epigallocatechin (EGC), and their derivatives acylated with gallic acid (ECG and EGCG). In relation to how the connection between the various units takes place, it is possible to distinguish proanthocyanidins of the A, B, and C series (Figure 1). The first are certainly less common; they are characterized by the presence of two catechin molecules joined together by a double bond, a C4→C6/8 carbon–carbon bond, and a hetero C2→O7 bond. Since on the C ring, the substituents can be in both S and R configuration, we can have four types of proanthocyanidins A (A1-A4). More common proanthocyanidins are the B series, which are formed from catechin or epicatechin through a single carbon–carbon bond C4→C6/8. Also in this case, in relation to the configuration of the various substituents on C2 and C3, many structures will be possible, which are indicated with B1-B4 and B5-B8 [27,28]. Proanthocyanidins of the “C” series have also been identified: these are trimers, in which three flavanols are joined by C4→C8 bond [29].

### 3.1. Bioavailability of Catechin and Their Polymeric Structures

The accumulation of data and scientific research suggests that diets rich in flavanols and procyanidins are beneficial for human health, yet, despite the large number of studies, there are still considerable differences and disagreements regarding the fate of metabolites of these compounds and their actual properties in vivo. Ottaviani et al. demonstrate that the beneficial effects on human health associated with the consumption of foods containing flavanols depend significantly on the stereochemical configuration of ingested flavanols; This is because stereochemical configuration has a profound influence on their absorption and metabolism in humans. As showed by Ottaviani et al., in vivo the biological activity of flavanols in the modulation of arterial function and vasodilation, regardless of the antioxidant properties shown in vitro, depends significantly on the stereochemical configuration of the compounds [30]. On the other hand, convincing clinical evidence highlight the flavanol-containing foods properties of inducing improvements in human vascular function and Alañón et al. indicated the intake levels of (−)-epicatechin as low as 0.5 mg/kg BW as full dose-dependency capable of inducing acute improvements in vascular function in healthy volunteers [31]. The efficiency of the multiple beneficial effects of flavonoids on human health is closely linked to their bioavailability, i.e., the speed and extent to which these compounds are absorbed and metabolized. The bioavailability of catechins, for instance, is influenced by several factors that include the type of food and the efficiency of the digestive process of everyone. In general, the absorption and metabolism of flavonoids follows a common pathway [32]. After ingestion, they are absorbed into the intestinal lumen and then pass into the circulatory system; a further metabolic process can take place in the liver. Some of them, not absorbed in the small intestine, pass to the colon where they undergo a biotransformation by the enzymes of the intestinal microbiota that leads to the production of phenolic compounds. However, their absorption is strictly conditioned by the size and hydrophobicity of the compound [33,34]. The compounds can be absorbed passively or by facilitated diffusion and are finally metabolized to sulphate and methylated forms. In the digestion process, PCAs with high molecular weight and degree of polymerization can interact with proteins to form less digestible complexes, while oligomeric PCAs are more rapidly absorbed [35]. In general, PCAs during digestion seem to be extensively metabolized by the intestinal microbiota mainly producing phenylacetic, phenylproprionic, and phenylbutyric acid [36]. In vivo and in vitro studies have shown that oligomeric PCAs are hydrolyzed to epicatechin in the rat small intestine [35]; Epicatechin and catechin are then further metabolized to methylated and glucuronidated conjugated, which forms predominate in the systemic circulation [37]. Polymeric PCAs with high molecular weight, on the other hand, show a permeability coefficient about 10 times lower than catechin and dimers, trimers of PCAs. Goodrich et al. from a study of luminal content after administration of grape seed extract to Sprague Dawley rats showed that procyanidins reach maximum concentration after about three hours while metabolites reach maximum concentration after about 3–18 h at the latest [38]. In human studies, after a consumption of (−)-epicatechin, oligomer, and polymeric proanthocynidin contained in a test drink or encapsulated in hard gelatin, γ-valerolactones but no dimeric or oligomeric PCAs were detected in the plasma, thus questioning the notion that PCAs are broken down into flavanols prior to their absorption [39,40]. A number of studies postulate that the bioavailability of catechins depends on the nutritional strategy and may decrease if accompanied by food [41,42]. In this context, a human study of Fernández et al. showed that EGCG administered in a single dose of 250 mg after overnight fasting, results in the highest plasma concentrations values, showing significant differences according to the conditions and nutritional supplements used [43]. In agreement with previous studies, these data would suggest an attenuated response of digestive processes following the administration of EGCG alone; the fast digestive process would reduce the EGCG molecules degradation that would remain stable [41]. Preclinical evidence has shown that catechin-rich green tea extract improves intestinal barrier function and reduces intestinal and systemic inflammation [44]. Furthermore, catechins contribute to alleviate diabetes by regulating hyperglycemia and improving insulin resistance [45]. 

### 3.2. Biochemical and Functional Aspects of Catechin and Their Polymers

In recent decades, catechin and their polymers have aroused considerable interest in the world of research because of their potential beneficial effects on human health. Based on their structure, catechin can interact with many enzymatic systems showing many biological activities. In literature, there is already clear evidence about their anti-inflammatory, anticarcinogenic, cardioprotective, antioxidant, antiviral, and antimicrobial activity [46]. These compounds are ubiquitous and present in many common foods, such as fruits and vegetables, and this makes them even more interesting. In fact, several epidemiological studies correlate the consumption of functional foods rich in flavonoids—the so-called “superfoods”—with a decrease in the onset of chronic diseases [47]. In general, the multitarget behavior of catechin, put in place by the interaction with many signaling pathways of our body, has opened the way to several studies on the “structure–activity relationship” (SAR) to better understand the relationship between chemical structure and biological activity of the various subgroups [48].

Malgorzata Latos-Brozio et al. investigated the correlation between chemical structure of catechin and their polymers with their anti-inflammatory and antioxidant power. Flavanols have a catecholic structure particularly rich in hydroxyl groups that are useful for scavenger activity. In detail, it has been demonstrated that the degree of flavan-3-ols polymerization increases antioxidant activity; therefore, proanthocyanidins and condensed tannins have a greater activity than monomers such as catechin; this is because the scavenging activity is influenced by the conjugation of 3-OH on the B ring of the various monomers. Further studies of structure activity relationship were carried out to understand which were the factors that best affected this type of antioxidant activity [49]. Among these factors, it has been observed the conjugation between the double bond in C2 and C3 and some donor electric groups, such as the carbonyl in position 4, modifies the dissociation constant of the hydroxyl groups, increasing the stability of the phenolics radical that is created on the ring [50]. The potential for reduction/oxidation of different flavonoids and flavanols as a function of the hydroxyls group’s location has been considered, and it has been observed that the antioxidant power is greater when the position is in 2′, 4′-di-OH, 4′-OH ≈ 3′, 4′-di-OH > 2, 3-double bond in conjugation with 4-carbonyl substitution, 3, 5-di-OH in conjugation with 4-carbonyl substitution, 3-OH in conjugation with 4-carbonyl substitution, 5-OH in conjugation with 4-carbonyl substitution, and 3, 5-diOH [51]. In addition, the mode of travel of these compounds within the bloodstream was investigated. From a biochemical/functional point of view, it has been observed that, thanks to their structure, flavanols interact with several molecules, including human serum albumin (HSA), the main blood protein present at systemic level [52]. This binding interaction has been observed by different spectroscopic techniques, such as UV-Visible spectroscopy, fluorescence spectroscopy, and circular dichroism (CD) analysis [53,54,55,56,57,58]. Trnkova et al. from acquired UV–vis spectra observed that the interaction of the compounds with HSA closely depends on the acylation on the C ring with gallic acid; in fact, the first experimental data have shown that, among all the compounds tested, catechin and epicatechin, although capable of binding, have a low binding constant with has [59]. When in subsequent experiments, the latter interacted with galloylated catechins such as catechin gallate, epicatechin gallate, gallocatechin gallate, and epigallocatechin gallate, there was an important variation in the absorption band (a slight reduction between 200–280 nm followed by a significant increase between 300–360 nm) and a shift to the right in the absorption spectrum. This net change shows the catechin gallate–HSA complex formation [60]. To better understand the site of interaction between HSA and galylated catechins, fluorescence studies were also performed, considering that HSA is a class B protein, whose fluorescence emission depends mainly on the Trp214 residue, present in Sudlow site 1. Also in this study, it was seen that adding increasing concentrations of galloylated flavanols to a known concentration of HSA, there was a gradual shutdown in the fluorescence emission of HSA. This evidence confirms not only that the formation of the complex takes place, but that this interaction is of type 1:1, and the binding occurs within the hydrophobic microenvironment around the Trp214 [61]. In addition, native electrophoresis studies have shown that increasing (−)-epigallocatechin gallate concentrations protect the oxidation of the HSA thiol group, due to their antioxidant activity [62]. Similar experiments to those on monomers have been carried out on PCAs to highlight the modalities of the bimolecular complex formation with HSA. This has been demonstrated by UV–visible spectroscopy and by fluorescence emission studies of HSA in the presence/absence of PCAs. In the presence of the latter, there is a fluorescence emission shutdown in proportion to the PCAs concentration increase and a shift of the absorption band toward the blue/left (from 336 nm to 316 nm). Only HSA tryptophan residues, and not tyrosine residues, participate in the interaction between the two compounds [63].

## 4. Catechins in Metabolic Syndrome

A growing body of evidence supports the key role of oxidative stress and inflammation in the development of MetS-associated comorbidities [64,65]. Obesity is one of the main risk factors inducing to the dysmetabolic state since it is characterized by a chronic inflammation state predisposing to the development of insulin resistance. In animal models and in humans, experimental studies indicate that adipose tissue becomes hypoxic and secretes adipokines because of fat mass expansion. This condition stimulates the colonization by inflammatory cells including macrophages, the main sources of ROS [66,67]. Oxidative stress may be one of the triggers of the onset and progression of this chronic disease. Systemic oxidative stress and inflammation are two closely related processes, but it is still difficult to establish the exact temporal sequence of their relationship in the progression of the metabolic state. Mitochondria are the main site of intracellular ROS production, but ROS and RNS can also result from numerous other oxidative metabolic activities including endoplasmic reticulum, peroxisomes, plasma membrane, and cytoplasm. Under physiological conditions, controlled production of free radicals such as superoxide ($O_{2}^{-}$), nitric oxide (NO∙), and nonradicals such as peroxynitrite (ONOOH), is important for the modulation of numerous metabolic processes and in cellular homeostasis as well as being fundamental in the immune system to inactivate viruses and inhibit bacterial growth [68]. However, the high reactivity of ROS and RNS makes them highly harmful when produced in excess compared to cellular antioxidant defenses because they will tend to react with proteins, lipids and DNA causing severe cell oxidative damage. High ROS production can induce irreversible damage affecting regulatory enzymes where the modification of the redox state triggers an alteration of cell signaling, this alteration may involve apoptotic pathways and/or uncontrolled tissue growth and induce cancer. In detail, increased ROS generation may also trigger proinflammatory signaling pathways provoking the release of some inflammatory mediators and transcription factors, such as nuclear factor kappa B (NK-kB) [68]. NK-kB is normally localized in the cytoplasm kept out of the nucleus through direct binding to a specific inhibitor (called IkB). Under oxidative stress conditions, activation of specific proteases cause the detachment of the IkB inhibitor and NF-kB translocation into the nucleus where it activates the immune system responses [61,62]. Oliveira-Marques et al. highlighted the modulatory action of hydrogen peroxide (H_2_O_2_) in NF-kB pathway [69]. NF-κB induces the expression of various proinflammatory genes, including those encoding tumor necrosis factor a (TNFa), interleukin-1,6,8 (IL-1, IL-6, IL-8), cyclooxygenase-2 (COX-2), inducible nitric oxide synthase (iNOS), and other proinflammatory cytokines. This inflammatory network also promotes the activation of NF-κB, so NF-κB and inflammation constitute a positive feedback loop that induces cellular DNA damage and promotes cell proliferation and transformation. In addition, recent research suggests a role of NFkB in inflammasome regulation [70,71]; increased ROS production is another sensor able of stimulating and activating the NLRP3 inflammasome which in turn induces the activation of caspase 1 and promotes the activation of prointerleukin-1β and 18 (pro-IL-1β, pro-IL-18) into their active forms [72]. In addition, in obesity fatty acids from adipose tissue can help activate components of NADPH oxidase and other metabolic oxidases, facilitating the generation of reactive species [73]. This metabolic state of alteration involves various cell types, such as hepatocytes, myocytes, adipocytes, endothelium, and immune cells triggering different responses that lead to an overall severe dysmetabolic state.

A growing body of evidence suggests that the flavanols present in green tea may have a positive influence in several processes involved in the onset and progression of MetS (Figure 2). Flavanols are natural antioxidants that can relieve oxidative stress through a direct or indirect action, such as eliminating free radicals, chelating metal ions, and improving antioxidant enzyme activity. The main target of catechins and PCAs are resumed in Table 1. Pereira et al. have shown that (+)-catechin has the antioxidant capacity in a dose-dependent way between 0 and 100 μM [74]. Several studies have shown an increase in the expression of the sodium gene induced by flavanols [75,76]. In mice, 0.2% of flavanols administration showed a significant activity increase of the main enzymes involved in antioxidant defense including SOD, CAT, and GSH; in human Hep G2 cells EC and ECG increased the activity of GPx and glutathione reductase [77,78]. In rats with acetic acid induced colitis, Ran et al. showed increased SOD activity in mice treated with EGCG compared to untreated mice [79]. Chiou et al. demonstrated, using flow cytometry, that catechins such as ECG can increase the amount of GSH in LPS-induced RAW 264.7 cells [80]. Catechins antioxidant properties are related to the phenolic structure that allows them to chelate different transition metals and/or donate electrons stabilizing radical species. In detail, the molecules can bind iron preferentially to the 3-hydroxyl-4-carbonyl group, then to the 4-carbonyl-5-hydroxyl group and to the C3′-C4′ hydroxyl if present [81]. The chelating property of catechins reduces the generation of dangerous radical species because the presence of free metal transitions through Fenton and Haber–Weiss reactions induces the formation of high-reactive hydroxyl radical. In addition, the presence of hydroxyl groups in the phenolic structure makes them able to stabilize the free radicals, via the donation of an electron/hydrogen atom, and in this way stopping radical chain propagation such as lipid peroxidation. The gallate group present in some catechins further enhances the antioxidant properties of compounds as it is a metal binding site [82]; in EGCG the metal chelating groups able to neutralize ferric ions are the 3,4-dihydroxyl substituents and gallic acid in the B ring [83].

Mitochondria are rich in metal-containing proteins and are also the main source of ROS [89,105]. Several studies suggest a key role of mitochondrial dysfunction in several disease states related to the onset and progression of MetS, including insulin resistance, high blood glucose levels, and diabetes mellitus. In fact, the mitochondrial function is essential to regulate an adequate release of insulin form the pancreatic cells [84,106]. In cell cultures, an increase in mitochondrial content was detected after exposure to 3 μM of EC, suggesting a beneficial effect of the molecule on mitochondrial biogenesis [107]; in addition, incubating isolated mitochondria with concentrations < 0.8 μM of EC resulted in better preservation of membrane integrity [108]. Overall, the data seem to indicate a direct effect of EC on the components of the mitochondrial respiratory chain. EGCG also has well-known beneficial properties as a preventing mitochondrial deterioration [109]; Kumar et al. showed a beneficial action of EGCG on mitochondria—in detail, 20 mg/kg EGCG significantly attenuated oxidative damage, mitochondrial, and striatal alteration in male Wistar rats treated with 3-nitropropionic acid [85]. These properties are confirmed by further studies on Alzheimer’s disease (AD) showing that EGCG can restore potential mitochondrial membrane damage and ROS production by reducing AD-induced mitochondrial damage [86]. The beneficial effects of EGCG in mitigating oxidative stress is also related to its ability to activate the Nrf2/ARE pathway at multiple stages. Nuclear factorE2 related factor 2 (Nrf2) is a transcription factor that is activated in response to inflammation and oxidative stress. In normal conditions, Nrf2 lies in cytoplasm bound to Keap1, increasing oxidative stress leads to its dissociation from Keap1 and translocation to the nucleus. In the nucleus, Nrf2 binds to ARE, which leads to cytoprotective antioxidant gene transcription (Table 1). Therefore, EGCG, by downregulating Keap1 and disrupting the Nrf2–Keap1 interaction, increases the nuclear Nrf2 level and improves cellular antioxidant and anti-inflammatory activity [110]. In addition, NADPH oxidase (NOX) is the main source of superoxide and H_2_O_2_ needed for defense against pathogenic bacteria and fungi in phagocytes. Excessive production of these reactive species produces oxidative stress. EC and its metabolites are able to reduce the catalytic activity of the enzyme by inhibiting the overexpression and activity of redox-sensitive IKK/NFkB, JNK1/2, and protein-tyrosine phosphatase1B (PTP1B) signals in mice [111,112]. PTP 1B is a tyrosine phosphatase that reverses the IRS-1 phosphorylation of Tyr residues to prevent insulin signaling; therefore, its inhibition can beneficially affect the insulin resistance pathology [90,113]. IKK and JNK induce phosphorylation of Ser/Thr residues of IRS-1, an event that blocks Tyr phosphorylation of IRS and inhibits insulin signal transmission; these changes may be reversed by EC supplementation [87,114,115]. Catechins can also inhibit NFkB activation and translocation at different levels [91,95,116]. Hsieh et al. demonstrate that, in PANC-1 cells, EGCG treatment reduces Akt phosphorylation/activation [117]. Akt regulates NFkB transcriptional activity by inducing phosphorylation and subsequent degradation of the IkB inhibitor; therefore, its inhibition blocks NFkB in the cytoplasm, reducing its proinflammatory action. Wang, et al. reported that EGCG treatment is inversely associated with cardiovascular disease risk [92]. The cardioprotective action of catechins involves the activity of eNOS [94]. There are two forms of eNOS: inactive form is bound to caveolin 1 (Cav-1) on the cytosolic side of the membrane, whereas the active form is released following an increase in intracellular Ca^2+^ [93]. eNOS is also finely modulated by many post-transcriptional modifications, including phosphorylation of Ser 1177 and 633 and Thr 495 residues. Ramirez-Sanchez et al. have demonstrated that treatment with epicatechin stimulated the cell membrane dissociation of eNOS in human coronary artery endothelial cells in cultures suggesting the presence of a cell surface effector with high specificity for the epicatechin stereoisomer [88]. The receptor on the membrane appears to be highly specific because the effect of catechin (same structure but different orientation of one of its rings) is much lower.

Kurita et al. showed that EGCG induces phosphorylation of protein kinase B (AKT) and eNOS in cultured coronary artery endothelial cells. It has been shown that all of the hydroxyl group of the gallate ring is essential for PI3-kinase/AKT dependent phosphorylation. The PI3-kinase/AKT pathway mediates phosphorylation of Ser 1177 on eNOS, leading to an increased NO production and vascular relaxation [118]. Ramirez-Sanchez et al. also demonstrated that EC treatment induced significant increases in IP3 and increased phosphorylation/activation of IP3R, CAMKII at CAMI, using cultures of HCAEC. These findings suggest that EC-induced eNOS activation is mediated by Ca^2+^/CaMI/CaMKII [118]. Flavanols have a beneficial action as antidiabetics and help to compensate for the negative effects caused by insulin resistance (Figure 3). Catechins reduce plasma glucose levels in the blood improving the signal transduction pathway that promotes glucose transport in cells via GLUT4 [119]. In particular, EGCG promotes the externalization of GLUT4 through the PI3/AKT pathway [96,102]. In addition, studies on nonobese type 2 diabetic Goto–Kakizaki rats showed that EGCG improved glucose homeostasis through down-regulation of the ROS-ERK/KNK-p53 pathway [120]. In isolated hepatocytes, EGCG inhibits glucose production through suppression of hepatic gluconeogenesis and activation of 5′-AMP-activated protein kinase (AMPK) [98,103]. Liu et al. demonstrated that IRβ is the receptor for EC; EC upregulates receptor expression both in vivo and in vitro. Therefore, EC contributes to the regulation of glucose metabolism through IRS-1/PI3K/Akt by acting on IRβ [121].

## 5. Proanthocyanidins in Mets

To date, many aspects related to the biosynthesis, distribution, and role of proanthocyanidins are still poorly understood. However, their antioxidant and anti-inflammatory properties have been demonstrated [122]. Several research studies have highlighted scavenger activity against radical species, including superoxide anion and hydroxyl ions; Nazima et al. demonstrated that PACs protect against cadmium-induced ROS production and lipid peroxidation in erythrocytes and rat lymphocytes [123]. In addition, PCAs stimulate the cell’s antioxidant defenses by upregulating key enzymes such as SOD, CAT, GPx, and hemeoxygenase-1 (HO-1) [99,124,125]. PCAs act directly on the MAPK kinase pathway (activated by stress), improving the expression and activity of antioxidant enzymes via ERK, JNK, and p38MAPK in culture of liver and rat cells [126]. El-Alfy et al. showed that PCAs improved tissue damage produced by type 2 diabetes by increasing GSH levels and reducing lipid peroxidation and total nitrite and nitrate levels [100]. A high uptake of PCAs has been associated with a reduced risk of diabetes; Sapwarobol et al. in a study on healthy subjects showed that supplementation with grape seed PACs (100 mg, 300 mg) reduced postprandial glucose after 15 and 30 min compared to a high-carbohydrate diet [101]. PCAs also contribute to the regulation of glucose homeostasis by inhibiting the absorption of intestinal glucose through a direct action on GLUT 1 and 2, the main glucose transporters in enterocytes [104]. In addition, it has been shown that PCAs reduce blood glucose levels in a dose-dependent manner favoring the expression and externalization of GLUT4 on the cell surface and improve insulin resistance by directly activating IR and other targets of the insulin signaling pathway [97,127]. Zhang et al. demonstrated that insulin-resistant HepG2 cells treated with PAC grape seeds (6.25 mg/mL) showed increased glucose uptake compared to control cells [128]. PCAs are also able to inhibit some digestive enzymes even more than anthocyanins; Han et al. demonstrated through molecular docking analysis that proanthocyanidin B2 interacts with several amino acid residues of α-glucosidase, inducing an inhibitory effect that attenuates postprandial blood glucose [129]. The interaction of PAC on digestive enzymes seems to be closely dependent on their structure and degree of polymerization [130]. In addition, an in vivo study on Wistar rats fed a high-fat fructose diet enriched with grape seed PACs (100 mg/kg) shows activation of the PI3K/AKT pathway, suggesting an effect of PCAs on AMPK [131]. AMPK activation by PCAs, in turn, leads to significant downregulation of the gluconeogenic enzymes glucose 6-phosphatase and phosphoenolpyruvate carboxykinase [132,133]. Overall, all these actions contribute to improving insulin resistance and limiting the risk of hyperglycemia related to diabetic disease. Several studies also support the beneficial and vasorelaxant properties of PCAs, suggesting an important role of these compounds to manage cardiovascular and metabolic risk factors. DalBo et al. demonstrated that (0.1–100 μg/mL) of PCAs rich fraction from *Croton celtidifolius* barks have a dose-dependent vasorelaxant effect dependent involving eNOS activation upon the NO/cGMP pathway in combination with hyperpolarization due to activation of Ca^2+^-dependent K^+^ channels, in rat mesenteric arterial bed and isolated mesenteric artery [134]. Sankar et al. showed PCAs regulation of the impaired cholesterol metabolism through 3-hydroxy-3-methylglutaryl coenzyme A reductase (HMGR) level reduction [135]. In addition, Rigotti et al. showed that grape seed proanthocyanidins were able to regulate sirtuin 1 and 3 expressions, in embryonic kidney cells (HEK-293 cells) exposed to H_2_O_2_. Thus, PCAs contributed to prevent H_2_O_2_ -induced mitochondrial dysfunction and apoptosis, maintaining cell viability via SIRT 1 activation [136]. A summary of the data is depicted in Table 1.

## 6. Human Studies

Numerous studies support the hypothesis that consumption of natural extracts, containing catechins and their polymers, alleviates metabolic syndrome and related diseases [6,9,31,44,120,124,137]. In 2012, following several clinical trials, the European Food Safety Authority (EFSA) claimed that 10 g of high-flavanol dark chocolate containing 200 mg of cocoa flavanols consumed daily leads to beneficial effects on endothelial dysfunction, assessed via flow-mediated dilation (endothelium-dependent flow-mediated dilation, ED-FMD). In detail, this study evidence changing in the effects of catechins and their polymers, present in cocoa flavanols, according to the frequency of intake. In subjects who consumed a single intake, the effect on the endothelium was observed for only two hours, while subjects with a more frequent intake showed an acute effect on ED-FMD through a marked production of NO [138]. Other clinical studies, regarding cardiovascular diseases, support the benefits of these compounds on vascular endothelium and pressure. In this context, randomized clinical trials suggest that flavanols can reduce the risk of cardiovascular diseases related to high blood pressure through a positive influence on both systolic and diastolic blood pressure. Brown et al. demonstrated EGCG supplementation was able to reduce diastolic blood pressure (−2.68 mmHg, *p* = 0.014 vs. placebo) in overweight or obese male subjects aged 40–65 years who received 800 mg/day of EGCG or placebo for 8 weeks [139]. Similar results have been observed by Chatree et al. in a study on obese subjects who received 150 mg/day of EGCC or placebo for 8 weeks; they reported that EGCG reduced systolic blood pressure (−6.92 mmHg, *p* = 0.036 vs. baseline) and diastolic blood pressure (−2.00 mmHg, *p* = 0.044 vs. baseline). In addition, in these subjects was determined a significant reduction in plasma triglyceride levels (−29.58 mg/dL, *p* < 0.05 vs. baseline) [140]. As mentioned earlier, metabolic syndrome, in addition to being related to cardiovascular pathologies, can be triggered by type II diabetes and imbalances in glucose levels. In a study conducted by Hodges et al., in the context of a human dietary intervention study with catechin-rich green tea extracts (GTE) (890 mg catechins for 28 days), it was demonstrated that flavanol administration decreased fasting glucose and intestinal inflammation in healthy and MetS adults. Following this clinical study, it was observed that both in subjects with Mets and in healthy subjects, the administration of GTE did not result in significant changes in triglyceride levels or high-density lipoproteins (HDL), but it caused a rapid decrease in glucose levels, even in fasting conditions, reducing insulin-related intestinal inflammation. This suggests that the gut-level anti-inflammatory activities of GTE may improve glycemic control, with reductions in intestinal inflammation contributing to enhanced insulin sensitivity [141]. The preventive and protective activity of quercetin and pure epicatechin has also been compared; Dower et al., in a clinical study conducted on 37 healthy subjects, reported significant reductions in plasma insulin, both average and fasting (−1.46 mU/L, *p* = 0.03 compared to placebo), in healthy men and women after daily intake of 100 mg of epicatechin (two treatment periods lasting 4 weeks each, separated by a 4-week washout period). Furthermore, the same study observed that the flavanols present in cocoa, compared to quercetin, improve the functionality of pancreatic beta cells by increasing the activity of AKT/PI-3 kinase and ERK1/2, which appear to play a key role in the pathogenesis of insulin resistance [142]. In some cases, it has been observed that epicatechin is able to simultaneously lower both plasma glucose and insulin levels. Specifically, Esser et al. highlighted a significant reduction in plasma glucose (−0.06 ± 0.38 mmol/L, *p* = 0.03 compared to placebo) and plasma insulin (−0.80 ± 2.21, *p* = 0.02 compared to placebo) in (pre)hypertensive subjects treated with EC (100 mg/day) for 4 weeks. Additionally, this study also demonstrated the correlation between glucose levels and inflammation, as EC treatment also affected the inflammation state by inhibiting NF-κB, TNFα, and IL-1β [143]. In overweight and obese men, Brown et al. observed a marked increase in the concentration of plasma EGCG, urinary EGC, and urinary 4′-O-methyl EGC after the assumption of decaffeinated green tea extract containing about 400 mg total catechins twice daily. In addition, during the intervention period, body weight decreased by 0·64 (sd 2.2) kg, suggesting a protective effect of green tea catechins on weight gain [144]. In support of these effects, a clinical trial on women with central obesity highlighted weight reduction and changes of lipid profile and obesity-related hormone peptides after 12 weeks of high-dose EGCG treatment [145]. In details, green tea extract (EGCG) at a daily dosage of 856.8 mg improved weight loss, from 76.8 ± 11.3 kg to 75.7 ± 11.5 kg (*p* = 0.025), as well as decreased in BMI (*p* = 0.018) and waist circumference (*p* = 0.023); also, a consistent trend of decreased total cholesterol and LDL plasma levels and lower ghrelin levels and elevated adiponectin levels were detected [145]. These findings, further confirmed by other studies, stress the role of green tea flavanols in improving features of metabolic syndrome in obese patients [146,147,148,149].

## 7. Conclusions

The present review explained the mechanisms involved in the development of MetS, focusing on the biochemical and functional aspects of flavanols and their polymers by researching how flavanols can affect the various biochemical targets characterizing MetS. The results are encouraging and show a wide range of efficiency of the compounds concerning, above all, their antioxidant, and anti-inflammatory properties. Flavonoid can attenuate the alteration of several molecular targets and may contribute to the control of the main states of metabolic alteration predisposing to MetS, including insulin resistance, diabetes, obesity, and cardiovascular disease. However, further studies are still needed to better elucidate the flavanols mechanisms of action, in order to optimize the dose to be used for the potential treatment. In fact, often the efficacy of the dose is linked to drug concentrations difficult to reach in vivo, where it is also necessary to consider the absorption, transport, and availability of active metabolites. In this context, the interaction of flavanols with HSA seems particularly promising study. The protein is emerging as a versatile drug carrier for biomedical applications, increasing the pharmacokinetics and solubility of some compounds in vivo. HSA–drug complex favoring the distribution of flavanols and their bioavailability could help to improve and potentiality of their use and efficacy as therapeutic agents.

## Figures and Tables

**Figure 1 ijms-24-09228-f001:**
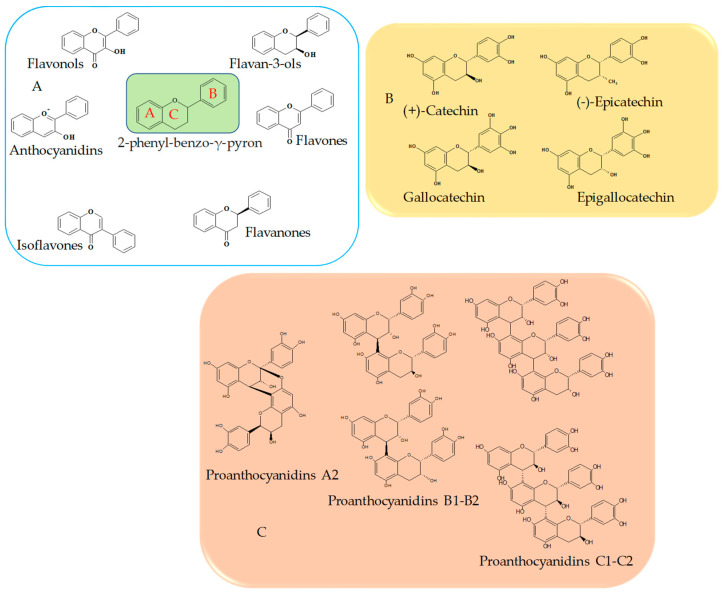
Schematic representation of the chemical structure of the treated compounds. (**A**) Basic skeleton of flavonoids and their main subclasses; (**B**) chemical structures of principal flavan-3-ols; (**C**) basic chemical structure of prohantocyanidin.

**Figure 2 ijms-24-09228-f002:**
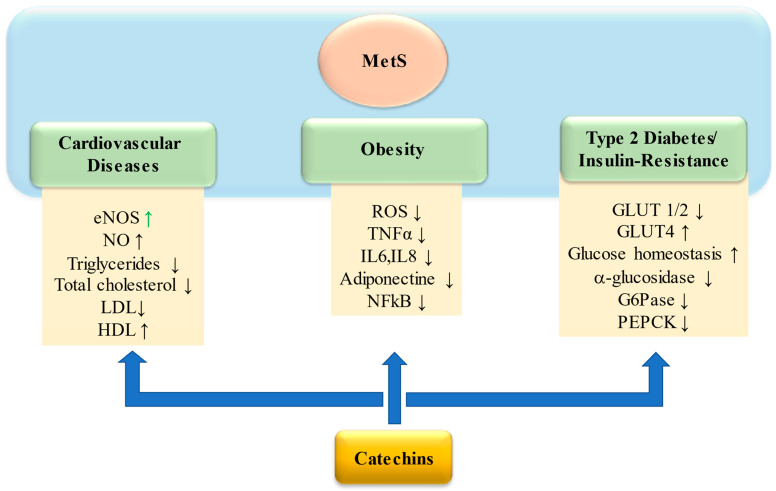
Beneficial effect of catechins in counteracting dysmetabolism by MetS. Catechins ameliorate cardiovascular flux: by modulation on eNOS that leads to increase NO production, which in turns improves vascular relaxation; by decreasing LDL, cholesterol, and triglycerides values that contribute to plaque buildup in arteries; by increasing HDL values. Catechins reduce inflammation process through reduction of ROS and NFkB, both triggering proinflammatory mediators including TNFa, IL-6, and IL-8. Catechins positively affect glucose homeostasis improving the glucose transport in cells via 4 and suppressing hepatic gluconeogenesis through G6Pase and PEPCK inhibition. Furthermore, catechins attenuate postprandial glycemia reducing GLUT 1 and 2 genes. Abbreviations: Nitric oxide endothelial synthase (eNOS); Nitric oxide (NO); Low-density lipoprotein (LDL); High-density lipoprotein (HDL); Reactive oxygen species (ROS); Tumor necrosis factor a (TNFα); Interleukin-6 (IL-6); Interleukin-8 (IL-8); Nuclear Factor kappa B (NK-kB); Glucose transporter (GLUT); Glucose 6-phosphatase (G6Pase); Phosphoenolpyruvate carboxykinase (PEPCK).

**Figure 3 ijms-24-09228-f003:**
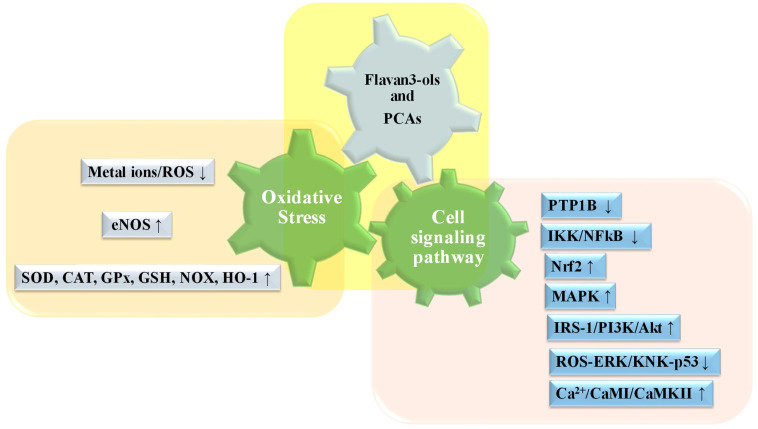
Schematic representation of the main molecular targets of catechins and PCAs. In general, they ameliorated MetS and related diseases improving cell antioxidant defenses while metal ions and ROS levels are reduced. MAPK, IRS-1/PI3K/Akt and Ca^2+^/CaMI/CaMKII pathways are improved while PTP1B, IKK/NFkB and ROS-ERK/KNK-p53 pathways are down-regulated.

**Table 1 ijms-24-09228-t001:** In vitro and in vivo studies on catechins and PCAs-mediated health effects.

Flavanols	Molecular Targets	Cell Culture	Dose	References
**EC**	Stimulated mitochondrial biogenesis	Mouse skeletal muscle	3–10 μM	[80]
	Reduced release of the cytochrome c	Isolated rat heart mitochondria	0.23 µg/mL and 0.46 µg/mL	[81]
	Decreased NOX3/NOX4 liver expression and mitigated oxidative stress	C57BL/6J mice	0 mg EC/kg	[84]
	Modulated NOX subunit expression and directly inhibit NOX. Mitigated HFr-induced insulin resistance	Rat model	20 mg EC/kg	[85,86]
	Stimulated the cell membrane dissociation and activation of eNOS	Human coronary artery endothelial cells	1 μmol/L	[87]
	Regulated glucose metabolism through IRβ receptor and IRS-1/PI3K/Akt pathway	IR-HepG2 cells	0–250 μg/mL	[88]
**EGCG**	Attenuated oxidative damage	Male Wistar rats	20 mg/kg	[83]
	Enhanced the expression of SOD	Mice	6.9 mg/kg	[67]
	Increased the gene expression of SOD1,2, CAT, and GPx	IMR90 (HDF) cells	25–50 μM	[68]
	Increased SOD activity	Male rats	50 mg/kg/d	[71]
	Activated the Nrf2/ARE signaling pathway	Normal Rat Kidney Epithelial Cells (NRK-52E) cell	5 μM	[89]
	Reduced Akt phosphorylation/activation	Human pancreatic cancer cell line PANC-1	0 to 20 μM	[90]
	Promoted GLUT4 translocation via activation of PI3K/Akt signaling pathway	Rat skeletal muscle L6 cell	50 μM	[91]
	Downregulated the ROS-ERK/JNK-p53 pathway and improved glucose homeostasis	Goto–Kakizaki (GK) rat	100 mg/kg/d	[92]
	Suppressed hepatic gluconeogenesis through 5′-AMP-activated protein kinase	Isolated hepatocytes	≤1 μm	[93]
	Inhibited glucose production	H4IIE rat hepatoma cells	25 μM	[94]
**EGCG + EGC**	Reduced intestinal SGLT-1/GLUT2 ratio and enhanced adipose GLUT4	Male Wistar rats	4 mg EGCG+2 mg EGC	[95]
**ECG**	Attenuated the ROS level	RAW 264.7 cells	10 μM	[72]
**PCAs**	Upregulated SOD, CAT, GPx, and hemeoxygenase-1 (HO-1)	Kunming Mice Liver	100 mg/kg	[96]
	Activated the NO/cGMP pathway	Rat mesenteric arterial	0.1–100 μg/mL	[97]
	Increased GSH levels and reduced lipid peroxidation and total nitrite and nitrate levels	Wistar rats	50 and 100 mg kg^−1^	[98]
	Decreased blood glucose level, increased insulin level through regulation of the PI3K signaling pathway	Male albino rats	300 mg/kg/day	[99]
	Activated the PI3K/AKT pathway	Albino Wistar rats	100 mg/kg	[100]
	Activated AMPK, Sirt1, and PGC-1α	Male BALB/c mice	100 and 200 mg/kg	[101]
	Activated the Nrf2 pathway	Wistar rats	250 mg/kg	[102]
**Procyanidins**	Reduced ROS formation, increased SOD, CAT, GPx activity; regulated MAPK kinase pathway	Sprague-Dawley (SD) rat	10–50 mg/kg	[103]
	Reduced the blood glucose level, PEPCK, and G6Pase	C57BL/6 male mice	10 mg/kg	[104]

Abbreviations: (−)-Epicatechin (EC); epigallocatechin gallate (EGCG); epigallocatechin (EGC); proanthocyanidins (PCAs).

## Data Availability

Not applicable.

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
