# Peer review of "Catechins and Proanthocyanidins Involvement in Metabolic Syndrome"

_ijms, 2023, doi:10.3390/ijms24119228_

Round 1
Reviewer 1 Report
The review by G. Tancredi Patanè et al. “Catechins and proanthocyanidins involment in metabolic syndrome” aims to selectively evaluate the benefits of catechins and their polymeric structure on metabolic syndrome. The research was properly conducted and adequately summarized.
Some points should be addressed before the acceptance
Figure 1 should be enlarged and moved to near line 102, where it is first named.
Table 1 should have a footer explaining the meaning of EC, EGCG, ECG, PCAs …
The font of all paragraphs should be unique, as specified in the template.
The 2, 20, 32, 33, 35, 39, 44, 45, 46, 57, 60, 80, 82, 101, 105, 110, 125, references must all be correct.
Anyway, a check on all references is recommended
Author Response
The review by G. Tancredi Patanè et al. “Catechins and proanthocyanidins involment in metabolic syndrome” aims to selectively evaluate the benefits of catechins and their polymeric structure on metabolic syndrome. The research was properly conducted and adequately summarized.
Some points should be addressed before the acceptance.
Figure 1 should be enlarged and moved to near line 102, where it is first named.
According to reviewer suggestion, Figure 1 has been corrected.
Table 1 should have a footer explaining the meaning of EC, EGCG, ECG, PCAs …
According to reviewer suggestion, the meaning of EC, EGCG, ECG, PCAs … has been explain in the footer of the table.
The font of all paragraphs should be unique, as specified in the template.
According to reviewer suggestion, the review has been entirely revised.
The 2, 20, 32, 33, 35, 39, 44, 45, 46, 57, 60, 80, 82, 101, 105, 110, 125, references must all be correct.
According to reviewer suggestion,, all references have been corrected.
Reviewer 2 Report
Several suggestions for further improvement of the quality of this review article are as follows,
1. The quality of Figure 1 should be improved.
2. Section 3. Chemical classification, this subtitle should be improved, and what kinds of compounds involve in classification. The logical of this section should also be improved.
3. The quality of Figure 2 is extremely poor, e.g., full name of all abbreviations should be added in the figure caption, and the potential mechanism of action of catechins could be summarized.
4. Table 1 should be improved, the logical of models should be cell studies, then animal studies.
5. The quality of Figure 3 is also extremely poor.
Author Response
- The quality of Figure 1 should be improved.
According to reviewer suggestion, Figure 1 has been improved.
- Section 3. Chemical classification, this subtitle should be improved, and what kinds of compounds involve in classification. The logical of this section should also be improved.
According to reviewer suggestion, the subtitle has been improved.
- The quality of Figure 2 is extremely poor, e.g., full name of all abbreviations should be added in the figure caption, and the potential mechanism of action of catechins could be summarized.
According to reviewer suggestion, the quality of the figure has been improved and the potential mechanism discussed
- Table 1 should be improved, the logical of models should be cell studies, then animal studies.
According to the suggestions, Table 1 has been revised.
- The quality of Figure 3 is also extremely poor.
All the Figures have been improved.
Reviewer 3 Report
Line 37: please add what these epidemiological studies are.
In general, authors overuse references to reviews in the introduction. I recommend that they add bibliography on clinical studies on flavonoids when talking about their effect on metabolism.
Line 57: add the WHO citation
Figure 1: the names of the structures are in Italian. Please revise
Line 111: the authors indicate "8 subgroups" but only cite six.
Line 115: the "o" in the word "categories" is in bold, please revise.
Regarding the bioavailability section, I would recommend that the authors add a human clinical study on the bioavailability of catechins in blood. In line 172 they talk about a study in rats but they could add studies of bioavailability catechins in humans and that this bioavailability depends on the nutritional strategy.
Line 302: the paragraph is cut off with figure 2, review
Figure 2: there are words in Italian. Please check.
Line 329: micromolar is given as µM, not µm (this would be micrometres). Revise this throughout the text. Also, commas should be replaced by full stops when decimals are used (e.g. line 331).
Line 337: the authors say "studies on AD", but what is AD?
Line 345: the table 1 you indicate does not relate to the information in table 1.
Point 5 should relate, as its title indicates, catechins to metabolic syndrome, but it mainly talks about their antioxidant and anti-inflammatory effect, repeating information from the previous point. I understand that metabolic syndrome is directly related to oxidative stress and inflammation but then points 4 and 5 should be merged. In this point 5, and after reading its title, I would expect information focused on studies in humans with metabolic syndrome or animals after taking catechins.
Line 459: the authors state "Flavonoid can attenuate the dysbiosis of several molecular targets". Dysbiosis is a term related to microbiota, please clarify this point.
Overall, the authors explain, in a very detailed way, the effect of catechins and protoanthocyanidins on key points of the metabolic syndrome, such as inflammation and inflammatory stress. However, as the abstract states, the aim of the work is "... to evaluate the effect of catechins and protoanthocyanidins on the metabolic syndrome.
The present review aims to selectively evaluate the benefits of catechins and their polymeric structure on metabolic syndrome, a common disorder characterised by a cluster of three main risk factors: obesity, hypertension, hyperglycemia". At the end of the introduction they again state "
Since there are different subgroups of flavonoids with different biological activities depending on the chemical structure and substituents, the present review aims to selectively evaluate the benefits of some categories of flavanol and their polymeric structure on MetS".
I was expecting to find more information on the use of these compounds in humans with metabolic syndrome, this information is very scarce and scattered in the text, with practically no references on the effect of these antioxidants and visceral fat, key in this syndrome.
I suggest the authors add a small section focusing specifically on this point as it is part of their objectives.
Author Response
Line 37: please add what these epidemiological studies are.
In general, authors overuse references to reviews in the introduction. I recommend that they add bibliography on clinical studies on flavonoids when talking about their effect on metabolism.
According to reviewer suggestion, the reference number “[4]” has been added.
Line 57: add the WHO citation.
Citation has been added.
Figure 1: the names of the structures are in Italian. Please revise.
According to reviewer suggestion, Figure 1 has been revised and corrected.
Figure 2: there are words in Italian. Please check.
According to reviewer suggestion, all the figures have been revised and corrected.
57: add the WHO citation
Citation has been introduced.
Line 111: the authors indicate "8 subgroups" but only cite six.
Line 111 has been corrected.
Line 115: the "o" in the word "categories" is in bold, please revise.
The world has been corrected.
Regarding the bioavailability section, I would recommend that the authors add a human clinical study on the bioavailability of catechins in blood. In line 172 they talk about a study in rats but they could add studies of bioavailability catechins in humans and that this bioavailability depends on the nutritional strategy.
According to reviewer suggestion, the bioavailability section has been enriched.
Line 302: the paragraph is cut off with figure 2, review.
According to reviewer suggestion, the figure has been moved.
Line 329: micromolar is given as µM, not µm (this would be micrometres). Revise this throughout the text. Also, commas should be replaced by full stops when decimals are used (e.g. line 331).
According to reviewer suggestion, the text has been entirely revised.
Line 337: the authors say "studies on AD", but what is AD?
List of abbreviations has been added and line 337 has been corrected.
Line 345: the table 1 you indicate does not relate to the information in table 1.
According to reviewer suggestion, in the sentence “Table 1” has been deleted.
Point 5 should relate, as its title indicates, catechins to metabolic syndrome, but it mainly talks about their antioxidant and anti-inflammatory effect, repeating information from the previous point. I understand that metabolic syndrome is directly related to oxidative stress and inflammation but then points 4 and 5 should be merged. In this point 5, and after reading its title, I would expect information focused on studies in humans with metabolic syndrome or animals after taking catechins.
According to reviewer suggestion, point 4 and 5 have been merged and improved.
Line 459: the authors state "Flavonoid can attenuate the dysbiosis of several molecular targets". Dysbiosis is a term related to microbiota, please clarify this point.
Overall, the authors explain, in a very detailed way, the effect of catechins and protoanthocyanidins on key points of the metabolic syndrome, such as inflammation and inflammatory stress. However, as the abstract states, the aim of the work is "... to evaluate the effect of catechins and protoanthocyanidins on the metabolic syndrome.
The present review aims to selectively evaluate the benefits of catechins and their polymeric structure on metabolic syndrome, a common disorder characterised by a cluster of three main risk factors: obesity, hypertension, hyperglycemia". At the end of the introduction, they again state "
Since there are different subgroups of flavonoids with different biological activities depending on the chemical structure and substituents, the present review aims to selectively evaluate the benefits of some categories of flavanol and their polymeric structure on MetS".
I was expecting to find more information on the use of these compounds in humans with metabolic syndrome, this information is very scarce and scattered in the text, with practically no references on the effect of these antioxidants and visceral fat, key in this syndrome.
I suggest the authors add a small section focusing specifically on this point as it is part of their objectives.
According to reviewer suggestion, a new point “Human studies” has been added.
Round 2
Reviewer 2 Report
The revised manuscript could be accepted.
Reviewer 3 Report
The authors have made a great effort to revise and improve the paper. Congratulations.